# Cryo EM structure of intact rotary H$^+$-ATPase/synthase from *Thermus thermophilus*

Atsuko Nakanishi[1], Jun-ichi Kishikawa[1], Masatada Tamakoshi[2], Kaoru Mitsuoka[3] & Ken Yokoyama[1]

Proton translocating rotary ATPases couple ATP hydrolysis/synthesis, which occurs in the soluble domain, with proton flow through the membrane domain via a rotation of the common central rotor complex against the surrounding peripheral stator apparatus. Here, we present a large data set of single particle cryo-electron micrograph images of the V/A type H$^+$-rotary ATPase from the bacterium *Thermus thermophilus*, enabling the identification of three rotational states based on the orientation of the rotor subunit. Using masked refinement and classification with signal subtractions, we obtain homogeneous reconstructions for the whole complexes and soluble V$_1$ domains. These reconstructions are of higher resolution than any EM map of intact rotary ATPase reported previously, providing a detailed molecular basis for how the rotary ATPase maintains structural integrity of the peripheral stator apparatus, and confirming the existence of a clear proton translocation path from both sides of the membrane.

[1] Department of Molecular Biosciences, Kyoto Sangyo University, Motoyama Kamigamo, Kita-ku, Kyoto 603-8555, Japan. [2] Department of Molecular Biology, Tokyo University of Pharmacy and Life Sciences, Horinouchi, Hachioji, Tokyo 192-0392, Japan. [3] Research Center for Ultra-High Voltage Electron Microscopy, Osaka University, 7-1, Mihogaoka, Ibaraki, Osaka 567-0047, Japan. Atsuko Nakanishi and Jun-ichi Kishikawa contributed equally to this work. Correspondence and requests for materials should be addressed to K.M. (email: kaorum@uhvem.osaka-u.ac.jp) or to K.Y. (email: yokoken@cc.kyoto-su.ac.jp)

The vacuolar type ATPases in eukaryotic cells (V-ATPase) mainly reside in membranes of intracellular compartments including the lysosome, endosome, and Golgi and function as proton pumps acidifying these compartments, a process essential for protein degradation, vesicle transport, and endocytosis[1–5]. The V-ATPase utilizes a rotary catalytic mechanism as seen in $F_oF_1$ ATP synthase, where the central rotor complex rotates relative to the surrounding stator apparatus (Supplementary Figure 1). Thus, these enzymes are termed rotary ATPases. These ATPases shares an overall architecture composed of a hydrophilic moiety ($F_1$/$V_1$) responsible for ATP synthesis or hydrolysis and a membrane embedded moiety responsible for proton translocation across the membrane ($F_o$/$V_o$)[1, 2]. Eukaryotic V-ATPases are likely to have evolved from homologous enzymes found in archea and some eubacteria[6–8], termed archaeal type ATPase (A-ATPase) or V/A type ATPase[8, 9]. The V/A ATPase from the thermophilic eubacterium *Thermus thermophilus* (*Tth*) is one of the best characterized rotary ATPases. The subunit composition of the *Tth*V/A-ATPase is similar to that of the eukaryotic enzyme but it has a simpler subunit structure and is responsible for ATP synthesis, using energy from an electrochemical potential generated by respiration to supply cells with ATP[10]. The hydrophilic $V_1$ part of the *Tth* enzyme is an ATP driven rotary motor in which the central DF shaft rotates inside a cylinder made of three alternately arranged A- and B-subunits[11]. The $V_o$ part of the *Tth*V/A-ATPase is composed of five different subunits; I, L, C, E, and G, homologous to eukaryotic a, c, d, E, and G, respectively. For clarity, in this paper subunit the terminology of the eukaryotic enzyme is used. The a-subunit is connected to the $A_3B_3$ stator in the $V_1$ domain by two peripheral EG stalks which form a peripheral stator apparatus ($A_3B_3E_2G_2a_1$). In addition, the dodecamer c ring composed of two transmembrane α-helices, the d-subunit and the DF shaft constitute the central rotor complex ($D_1F_1d_1c_{12}$), which rotates relative to the surrounding stator apparatus (Fig. 1a and Supplementary Figure 1). The proton motive force generated by respiratory complexes drives proton translocation through a proton pathway made up of both the a-subunit and the $c_{12}$ ring. This results in rotation of the whole central rotor complex which drives the cooperative synthesis of ATP from ADP and inorganic phosphate at the three catalytic sites in $A_3B_3$. Conversely, powering $V_1$ using the energy released by ATP hydrolysis, results in rotation of the DF shaft in the reverse direction. The tip of the DF shaft interacts with a funnel shaped d-subunit, driving rotation of the $c_{12}$ ring and resulting in proton translocation through $V_o$. According to a widely accepted model of $V_o$, a proton enters an access channel and binds to a glutamate on one of the c-subunits in the $c_{12}$ ring, following one revolution of the ring, the proton is released on the other side of the membrane via an exit channel[12]. In this model, the number of c-subunits in the ring is equal to the number of protons transported per revolution. Indeed, when powering $V_1$ by ATP, we have previously observed twelve steps for each rotation of the $c_{12}$ ring, a process assumed to be coupled to proton translocation[13]. This suggests an intrinsic flexibility of the *Tth*V/A-ATPase allowing accommodation of different gear sizes; (i) a three fold symmetric $V_1$ results in a 120° step rotation for each ATP hydrolysis and (ii) a 12-fold symmetric $V_o$ giving 30° step rotation for every proton translocation (Supplementary Figure 1).

Recent progress in single particle analysis of protein complexes using cryogenic electron microscopy (cryo-EM) has allowed structure determination of complete rotary ATPases revealing secondary structure level detail[8, 9, 14–17]. The first report of a cryo-EM structure of the $F_oF_1$ dimer by Davies et al. revealed that the $F_o$ a-subunit unexpectedly contained highly tilted transmembrane helices[14]. Nevertheless, the limited (6–7 Å) resolution of structures of the whole complex limited understanding of the molecular mechanism of rotary ATPases including both F- and V-type enzymes. Recently Schep et al.[9] revealed two rotational states of the *Tth*V/A-ATPase with different central rotor positions, using single particle analysis. The EM map of one state was reconstructed from the major data set of particle images to a limited resolution of 6.2 Å, insufficient for assignment of the

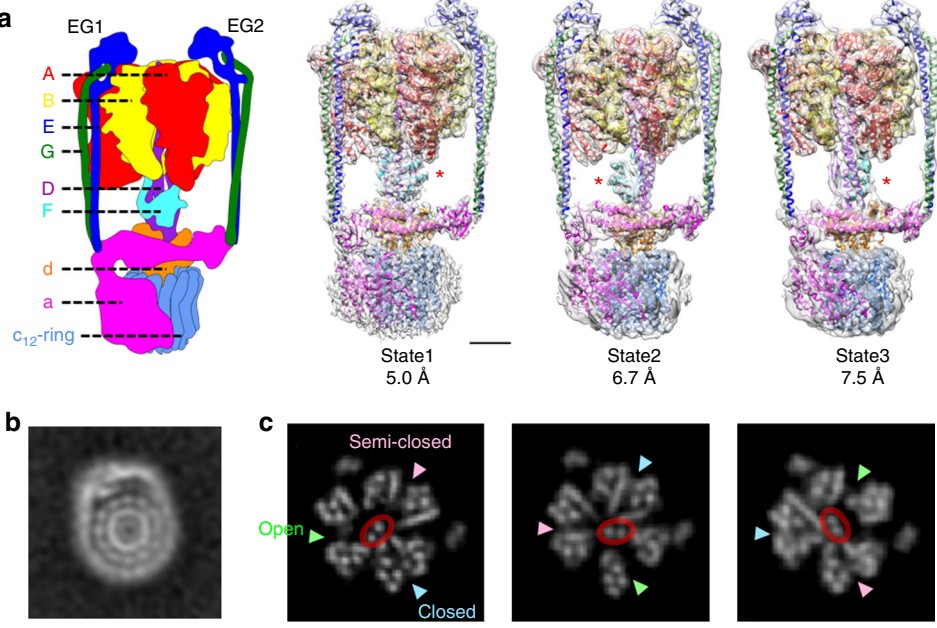

**Fig. 1** Three different structures of the V/A-ATPase from *T. thermophilus*. **a** A schematic model of V/A-ATPase from *T. thermophilus* is shown in left and the color of each subunit is corresponding to the models shown in right. Atomic models of the three rotational states were fitted into the maps by MDFF (right). The experimental maps are shown in semi-transparent gray. The central rotor subunit F is indicated by a red asterisk for each state. Scale bar, =30 Å. **b** A cross-section through the hydrophobic region in state1, showing that α-helices of outer ring of c-subunits are well resolved. **c** Cross-sections through the hydrophilic region of each map. The coiled-coil of the D-subunit is circled in red. The three AB interfaces are indicated by arrows in the section of state1

bound nucleotide at the catalytic sites in the frozen native state or for providing details of how each subunit interacts with partner molecules in the complex. The recent high resolution EM structure of yeast $V_o$ provided a near atomic resolution model of the membrane embedded region responsible for proton translocation[18]. This structure revealed an apparent aqueous cavity accessible from the cytoplasmic side and capable of proton translocation, although no periplasmic proton pathway could be clearly discerned in this structure. A periplasmic channel had been suggested in some earlier low resolution EM maps of rotary ATPases[8].

Here, we show cryo-EM maps corresponding to three different rotational states of the TthV/A-ATPase. The three different structures provide insights into the rotation of the whole complex allowing assignment of the movement of each subunit during rotation. The EM maps reconstructed from the major data set allowed us to construct a more detailed model of the TthV/A-ATPase than any reported previously. Furthermore, we analyzed both the whole complex and $V_1$ domains using masked refinement/classification[19], providing the detailed insight into the contact surface between rotor subunits and molecular basis of structural robustness of the peripheral stator apparatus. These results show the close integration of the different domains of rotary ATPase allowing tight energy coupling between the $V_1$ and $V_o$ moieties.

## Results

**Three different EM maps of T. thermophilus V/A-ATPase.** We used a cryo-EM Titan Krios (FEI) to obtain a high resolution EM map of the V/A-ATPase from the thermophilic eubacterium T. thermophilus. The V/A-ATPase in lauryl maltose-neopentyl glycol detergent (LMNG) at 0.003%, a concentration lower than the critical micelle concentration, was used for preparation of cryo-grids in order to obtain high contrast images[20] (Supplementary Figure 2).

In total we picked 483,129 individual particles of the TthV/A-ATPase from 4674 micrographs imaged by the direct electron detector CMOS camera (Falcon II, FEI). The strategy of single particle analysis for TthV/A-ATPase is summarized in Supplementary Figure 3. Following 2-D class averaging, a data set of 223,982 particle images was selected and subjected to 3-D classification analysis. The 12 classes obtained allowed identification of three distinct classes defined by the position of central subunit F. Then each data set was subjected to further classification using masked refinement/classification, as summarized in Supplementary Figure 3. After refinements, 57.9 % of the particles corresponded to state1 and gave maps at a resolution of 5.0 Å, higher than those of state2 at 6.7 Å resolution and state3 at 7.5 Å, containing 15.2% and 6.3% of the particles, respectively (Supplementary Table 1). Cross-sections through the maps show that α-helices are well resolved at all these resolutions (Fig. 1b, c).

The V/A-ATPase classes corresponding to the different rotational states were populated more unequally than that seen in EM studies of the yeast V-ATPase or bovine F type ATPase[15–17], and as seen in a similar study by Schep et al.[9]. The population of the class3 was less than ~ 7% of the total particles. This suggests that state3 of the TthV/A-ATPase represents the most energetically unstable structure of the ATPase. The molecular basis of this instability of state3 is discussed later.

The position of the central rotor DF-subunit in the three different states corresponding to each state is in good agreement with the 120° steps taken during the catalytic cycle of $V_1$. This is accompanied by re-arrangement of the AB pairs adopting different conformations in each state (Fig. 1c). Therefore, each map appears to correspond to a rotational state of the V/A-

ATPase. Available crystal structures and homology models were docked into the three maps and their conformations refined by molecular dynamics flexible fitting[21] (MDFF, Fig. 1). Alignment of an AB pair from each state shows that the C-terminal domain of the AB pair appears to move inward and outward accommodating movement of the central rotor DF critical for catalysis (Fig. 2 and Supplementary Movie 1 and 2).

**Nucleotide binding sites of the V/A-ATPase.** Local resolution analysis by Resmap[22] for state1 showed a range of resolution from ~ 4 Å in the $V_1$ region to ~ 5 Å in the $V_o$ domain, indicating conformational heterogeneity within the complex (Supplementary Figure 4). To overcome structural heterogeneity, we employed masked refinement with $V_1$EGd soft mask for the hydrophilic region. The density of the $V_o$ domain was blurred following this focused refinement. The resolution for the hydrophilic $A_3B_3DF(EG)_2d$ region was improved in both state1 and state2. (Fig. 2a and Supplementary Figure 3). For atomic model construction, crystal structures[23–27] and homology models were fitted into the three maps as rigid-bodies in Chimera[28] and the conformations refined by MDFF with secondary structure restraints. The map of state1 at 4.7 Å resolution produced a well defined atomic model by MDFF, particularly in the $V_1$ region (Supplementary Figure 4). The three AB pairs in $A_3B_3$ adopt three different conformations known as 'open', 'closed', and 'semi-closed', as seen in previously reported structures of the $F_1$ and $V_1$ complexes[25, 29, 30]. In this model, strong positive density was observed near the P-loop in two AB pairs; the 'closed', and 'semi-closed' state (Fig. 2). This is despite the fact that, the V/A-ATPase used to prepared cryo-EM grid was not supplemented with nucleotide or nucleotide analog. A molecule of ADP fits well into each density. There is no obvious density adjacent to the β-phosphate of the fitted ADP at each region, suggesting these densities represent bound ADP and not ATP. No density is observed near the P-loop in the AB pair in the 'open' state. Similar densities were also assigned as ADP in the map of state2 and state3 in the nucleotide binding sites, albeit the densities were less obvious in state3 (Supplementary Figure 5). Our previous study showed that the V/A-ATPase isolated from T. thermophilus membrane has no ATPase activity[31]. Thus, these results indicate that the V/A-ATPase used for the single particle analysis adopts the 2ADP bound form where both closed and semi-closed

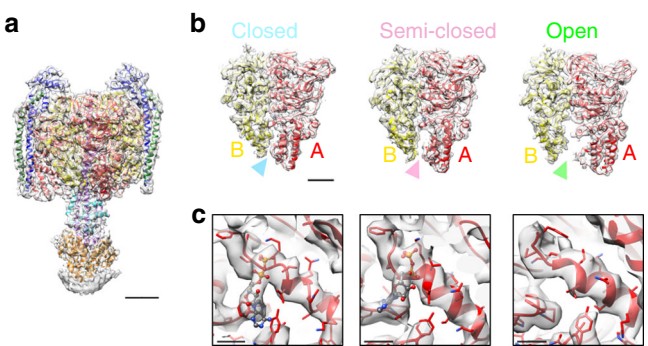

**Fig. 2** Structural comparison of the three AB pairs in state1. **a** The EM map in state1 with focus on the $V_1$EGd ($A_3B_3DF(EG_{CT})_2d$) subdomain. The density of the membrane embedded region was subtracted by focused refinement using the mask of $V_1$EGd. The map is shown in semi-transparent gray. Scale bar, =30 Å. **b** Side views of the three AB pairs, "closed", "semi-closed", and "open" corresponding to those of the cross-section shown in Fig. 1c. Scale bar, =20 Å. **c** Magnified views of the nucleotide-binding site at each AB pair. ADP is depicted in ball and sticks format. Experimental maps are shown in semi-transparent gray. Scale bar, =5 Å

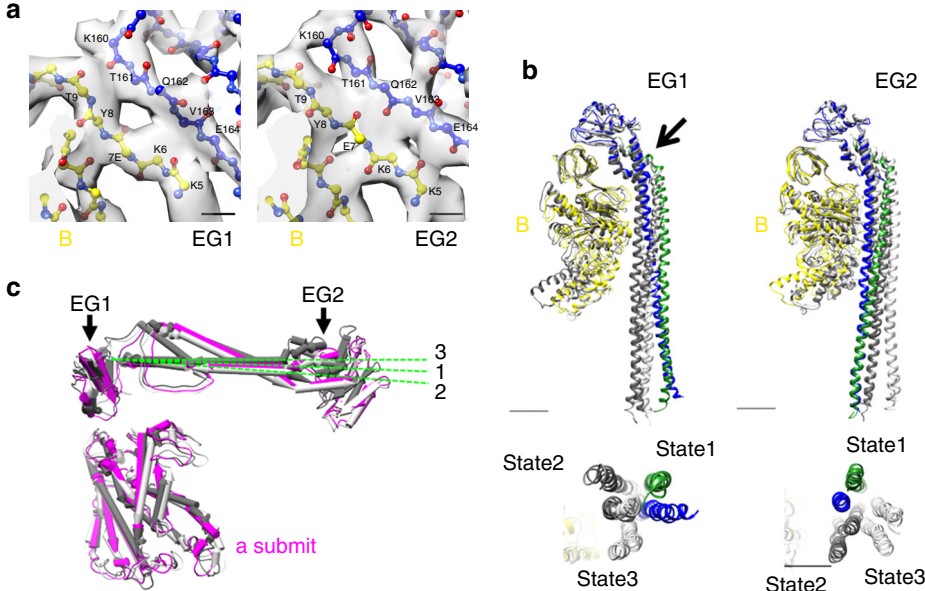

**Fig. 3** Structure of the peripheral stator subunits. **a** Close-up views of the interaction surface of B- and E-subunits in state1. Residues are depicted in sticks format and experimental maps are shown in semi-transparent gray. Scale bar, =3 Å. **b** Circular motions of EG1 and EG2 during transition of three rotational states. The E-subunit in state1 is colored blue, and the G-subunit is colored green. E- and G-subunits in state2 are shown in dark gray, and those in state3 are shown in light gray. These subunits are superimposed at the N-terminal β-barrel domain of the corresponding B-subunit. The arrow indicates the pivot point of the right handed coiled-coil of the EG1. The lower panels are bottom views of the N-terminal region of the EG-subunits. Scale bar, 20 Å. **c** Comparison of the N-terminal domain of a-subunit in three states. Subunits are superimposed at the C-terminal domain of the a-subunit. These subunits in state1–3 are colored in magenta, dark gray, and light gray, respectively

nucleotide binding sites are occupied by ADP, and thus corresponds to the auto-inhibited state known as the ADP inhibited form[32].

**Conformational transitions of peripheral stator**. The $V_1$ and the hydrophilic N-terminal region of the a-subunit (a-NT) are connected by two peripheral EG stalks (Fig. 1a). The EG bound to the N-terminal tip of a-NT is termed EG1 and the other stalk termed EG2, hereafter. Our improved EM map provides more detailed structural information than the previously reported map (see Supplementary Figure 6). Our higher resolution model of state1 reveals that there are close interactions between E/160–164 of E-subunits and B/5–9 of B-subunit, where rigid β-sheet structures are formed (Fig. 3a). This is consistent with our previous results showing that the $A_3B_3$ of the V/A-ATPase is tightly associated with the two EG stalks[33]. While the fitting of secondary structure elements to EM maps was reasonable in both state2 and state3, corresponding models showed clear conformational changes of the EG stalk. Superposition of the β-barrel domain of the B-subunits attached to the EG stalk reveals no apparent difference in the globular region composed of residues of E/99–188 between the three states (Fig. 3b). However, the stalk region of EG1 adopts significantly different conformations in the three different states. There is a pivot point of the right handed coiled-coil of the EG1 at both E/94–97 and G/104–107 in each state, resulting in a circular motion of the stalk. The conformation of EG2 in states1–3 also differs significantly, due to a similar circular motion (Fig. 3b).

The N-terminal tips of both EG1 and EG2 bind onto a-NT at different sites. Thus the motion of EG1 and EG2 during rotation induces movement of the a-NT. The a-NT in state1 has a different conformation to that in state2, but has a similar arrangement to that in state3 in the hydrophobic region of the a-subunit (Fig. 3c). The density for the binding surfaces of the tips of both EG stalks to the a-NT were less clear even in the map of

state1. The linker region of the a-subunit connecting the hydrophilic and hydrophobic regions also shows weak density in the map of state1, suggesting that the regions connecting the stalk region of EG and a-NT are highly flexible (Supplementary Movie 1). These comparisons clearly indicate that the stalk regions of both EG1 and EG2 are asymmetrical during rotation of the central rotor, and this is likely coupled to motion of the a-NT relative to the a-CT. These conformational changes during transition between the three rotational states can be seen most clearly in Supplementary Movie 3. The conformational changes in the stator subunits in the three rotational states have been reported for yeast V-ATPase[15]. We discuss the difference in stator movements during rotation of the eukaryotic V-ATPase and *Tth*V/A-ATPase later.

**Structure of central rotor**. Subunits D and F in the map are at relatively high resolution as shown by Resmap, allowing the generation of a well fitted model using the density of state1 (Fig. 4). At the secondary structure level, there was no apparent difference among the models of DF in the three states (Supplementary Figure 7a), suggesting that the coiled-coil region of the D-subunit is more rigid than previously thought[34]. In contrast, when superimposing the d-subunit in state1–3, the orientation of the C-terminal helix of the D-subunit in state3 changes drastically relative to that of state1 and state2 (Supplementary Figure 7), indicating state3 adopts a different conformation of rotor complex in order to accommodate distortion of the enzyme during rotation. In the yeast V-ATPase structures, the orientation of the C-terminal helix of the D-subunit in each state differs (ref. [15] and see Supplementary Figure 7).

The loop region at the tip of the D-subunit, missing in the crystal structure, is well assigned in our EM map of state1. At the $V_1$-DF/$V_o$-d boundary surface, it has been postulated that a short helix (D/74–81) at the tip of the D-subunit binds into the socket of the d-subunit, forming a sufficiently close interaction for

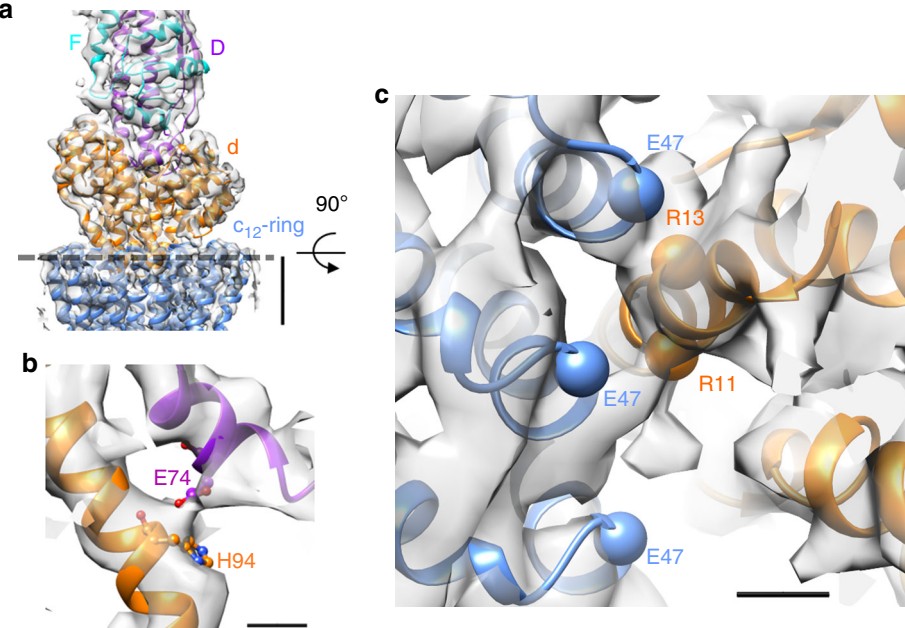

**Fig. 4** Structure of the central rotor. **a** A model of the center of the rotor complex. The d-, D, F, and c-subunits are colored in orange, purple, cyan, and blue, respectively. The experimental map is shown in semi-transparent gray. Scale bar, 20 Å. **b** A magnified view of the boundary surface of the central rotor between $V_1$-DF and $V_o$-d. Residues of E74 on the D-subunit and H94 on the d-subunit are ball and sticks format. Scale bar, =5 Å. **c** A magnified view of the boundary surface of the d-subunit and inner helices of the c-subunits. Residues of E47 on the c-subunits and R11 and R13 on the d-subunit are depicted in sphere formats. Scale bar, =5 Å

transmission of torque[35]. As previously reported, the d-subunit in the complex adopts a more open conformation than the crystal structure of d-subunit[36]. As a result, it was not possible to fit the full crystal structure into the EM maps effectively. Thus, we divided the d-subunit into three domains to fit individually into the EM density. Domain2 and 3 fit well into our EM maps but domain1, which includes N- and C-terminal regions, is rotated to open a space for the D-subunit binding. Thus, we did an additional rigid-body fitting of domain1 (18–77 and 282–323) and connected the loops to domain 2 and 3 and then MDFF fitting was performed for the entire model[37]. The obtained model fits well into the density (Fig. 4a), providing a detailed picture of the interaction surface between the short helix of the D-subunit and the d-subunit cavity. In this model, a loop region (D/115–120) and a short helix (D/74–81) of subunit D are inserted deeply into the cavity of the d-subunit, indicating close association of the two subunits. The E74 of the D-subunit appears to be in close proximity to H94 in the cavity of the d-subunit (Fig. 4b), indicating an interaction key for proper fitting of the short helix of the D-subunit into the d-subunit cavity. This is consistent with our previous results[35] demonstrating that exchange of the short helix including E74 of the D-subunit with the same equivalent regions from the enzyme of other species causes a complete loss of coupling between $V_1$ and $V_o$. Our improved EM map also reveals that the N-terminal helix of the d-subunit forms tight interactions with the two inner helices of the $c_{12}$ ring through electrostatic interactions; between d/R11 and c/E47 and d/R13 and c/E47 (Fig. 4c). This tight interaction between d- and $c_{12}$ ring is also responsible for the overall robustness of the rotor complex. A similar contact surface was reported for the $V_o$ of the yeast enzyme[18], suggesting evolutionary conservation of the rotor structure of V/A type ATPases.

**Structure of the membrane embedded domain**. The hydrophobic C-terminal region of the a-subunit (a-CT) and the $c_{12}$ ring

constitutes the hydrophobic domain responsible for proton translocation driven by the rotary mechanism (a-CT$c_{12}$). As described above, local resolution of the hydrophobic region, especially of the $c_{12}$ ring, was limited to ~6 Å due to conformational heterogeneity even within the most populated state1 structure. In order to improve resolution of the $V_o$ domain, we employed the focused classification by using signal subtraction of hydrophilic region for state1[19], as described in Supplementary Figure 3. The improved map of state1 at 5.0 Å resolution allowed us to build a model of the whole complex. Combining the crystal structure of the hydrophilic region of the a-subunit (3RRK)[38] and of the membrane embedded region, we could build an atomic model of almost the whole a-subunit within the overall complex. The main chain models of a-CT fitted well into the EM map, providing a more reliable model for the membrane domain of the whole complex than has previously been reported (Fig. 5 and Supplementary Figure 6).

It was possible to assign the helix bundle composed of three α-helices (Fig. 5a, a-TM1−3), three slightly tilted α-helices (Fig. 5a, a-TM4−6), and two highly tilted long α-helices (Fig. 5a, a-TM7 and 8). The inclined α-helices a-TM7 and a-TM8 are intimately associated with the $c_{12}$ ring composed of 12 c-subunits, each with two transmembrane α-helices joined by a short loop on the cytoplasmic side. The 12 outer α-helices of the c-subunit are clearly assigned, but the inner helices were not well resolved and form a concentric ring in the map (Fig. 1b and Supplementary Figure 6). The tilted a-TM7 contains the conserved R563, corresponding to R735 in the yeast enzyme, which together with E63 in the c-subunit forms an essential feature of proton translocation (Fig. 5b). The density for a-TM7, where the R563 is located, is in close proximity to the center of an outer helix of the c-subunit. The EM map of state1, without post processing of the EM map to reduce solvent density, revealed large pores in the detergent micelle on both the cytoplasmic and periplasmic sides of the protein (Fig. 5c–e). The cytoplasmic pore, previously reported in other EM structures of rotary ATPases[8, 9, 14–17],

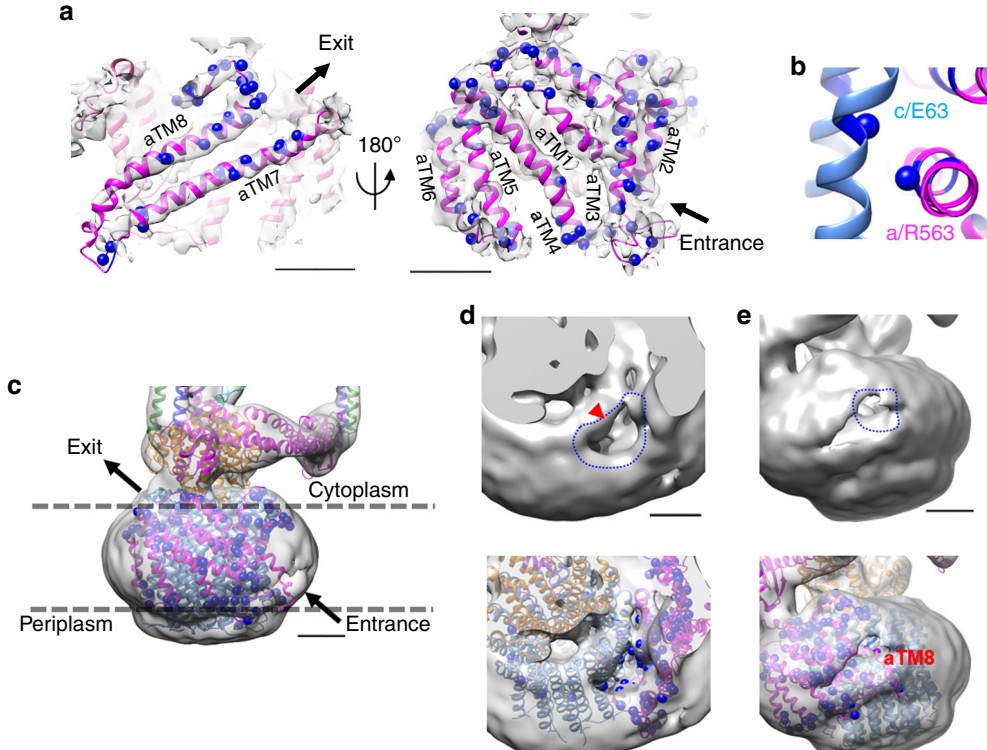

**Fig. 5** Structure of the membrane embedded domain. **a** The side views of the a-subunit. The homology model of the a-subunit was fitted into the map of state1 by MDFF. Expected paths of proton exit and entrance are shown by black arrow when the enzyme functions as an ATP synthase. Polar residues are depicted in blue sphere format. Scale bar, =20 Å. **b** A magnified view of contact surface between the c-ring and the a-subunit at cytoplasmic pore. The essential residues coupled protonation (E63 on c-subunit and R563 on a-subunit) are shown by sphere format. **c** Overview of the membrane embedded domain. The directions of the proton flow indicated by black arrows correspond to a. Scale bar, 20 Å. **d**, **e** Close-up views from the cytoplasmic (**d**) and periplasmic (**e**) side. Large cavities are outlined in blue. Viewed from the cytoplasm (**d**), density corresponding to residues indicated in **b** is apparent (red arrow). Scale bar, 20 Å

extends along the inclined a-TM7 and a-TM8, and ultimately exposes the interface between R563 in a-TM7 and the c-subunit (Fig. 5b) to the aqueous environment. On the periplasmic side, another cavity is more clearly observed in the detergent micelle than previously reported for other EM maps of intact rotary ATPases[8, 9, 14–17]. This cavity connects the narrow tunnel to the inclined a-TM8, indicating that both cavities are connected at the interface between the a-subunit and the $c_{12}$ ring. It is possible that these cavities identified in our EM map are the two hypothetical half-channels postulated on the basis of biochemical studies and mechanical analyses[12].

## Discussion

The cryo-EM map of the V/A type ATPase described here has revealed the most detailed model to date for any intact rotary ATPase. The model at a resolution of 4.7 Å for the hydrophilic region provides detailed structural information for the three nucleotide binding sites composed of the AB pairs, in the open, closed and semi-closed conformations. We also identified bound ADP in both the closed and semi-closed catalytic sites. The V/A-ATPase was frozen on the grid in the absence of additional nucleotide, thus the bound ADP has co-purified with the protein directly from the cell either in the form of ADP or is the result of ATP hydrolysis. The previously reported crystal structure of $Tth$V$_1$ at 3.9 Å resolution, also shows nucleotides occupying the two nucleotide binding sites, as a result of co-crystallization with ADP. Thus, the V$_1$ region of our cryo-EM map seems to represent the same catalytic state as the crystal structure of V$_1$. The

crystal structure of V$_1$ fits well into the cryo-EM density map at the secondary structure level, but the contact region of V$_1$ to both EG and d-subunit in the EM map differs slightly (Figs. 3 and 4).

In the crystal structure of the 3ADP bound form of *E. hirae* V$_1$ ($3_{ADP}$V$_1$)[39], electron density corresponding to sulfate, an analog of phosphate, was identified at the semi-closed nucleotide binding site, in close proximity to the β-phosphate group of ADP. No such density corresponding to phosphate was identified in either the closed and semi-closed sites of the *Tth* V/A-ATPase, although density for ADP was clearly identified at the two nucleotide binding sites. Unlike the *E. hirae* enzyme, the *Tth*V/A-ATPase is highly sensitive to ADP inhibition, where the entrapped ADP in one catalytic site inhibits further binding of ATP into other catalytic sites[30, 31]. In addition, our previous study indicated that the binding affinity of *Tth* V/A-ATPase for phosphate was much lower than that of both F$_1$-ATPase from various species and the *E. hirae* enzyme[32]. Thus, we conclude that the EM map presented here corresponds to the ADP inhibited state after phosphate has detached from the semi-closed site leaving ADP still bound. Indeed, the isolated V/A-ATPase from the *T. thermophilus* membranes shows very little ATPase activity until bound nucleotide is removed. It is noteworthy that the *Tth*V/A-ATPase functions as an ATP synthase, which gives an apparent pmf driven activity of ~60 ATPs/s[10]. The ADP inhibited form of V/A-ATPase is physiologically advantageous for cells as it prevents consumption of ATP when proton motive force is lost.

The structure of intact V/A-ATPase also provides a wealth of information on the ability of the rotary ATPase to resist mechanical torque generated by the motor domain. At the surface

of the V/A-ATPase central rotor, the d-subunit forms a socket-like structure that accommodates the DF shaft, indicating that the DF shaft in $V_1$ does not contact the $c_{12}$ ring directly[23]. Thus, interaction of the d-subunit with the D-subunit is key for torque transmission between the $V_1$ and $V_o$ domains. The model presented here provides insight into how the tip of the D-subunit interacts with the cavity of the d-subunit. The short helix (D/74–81), previously termed the driver helix[35], is inserted deeply into the cavity of the d-subunit, allowing formation of a key electrostatic interaction between the side chains of D/E74 and d/H94 in the cavity. In addition, the loop region (D/115–120), albeit of slightly lower resolution in our map, appears to also be in contact with the cavity of the d-subunit. The d-subunit in the complex adopts a more open form than in the crystal structure of the monomer, as indicated previously[36]. These associations at the surface of the two subunits likely induce conformational change of the d-subunit making it more open and thus able to accommodate the tip of D-subunit into the cavity. This association is also likely to contribute to the relative stiffness of the rotor complex.

The two peripheral EG stalks of the V/A-ATPase are also key for mechanical coupling of the proton motive force to the ATP synthesis in $V_1$ by connecting the $A_3B_3$ domain to the hydrophobic domain of the a-subunit in $V_o$. The present improved EM map provided detailed structural information on how the EG stators bind to the B-subunit. The main interactions between the EG stalks and the $V_1$ domain involve the E-subunit, which sits on top of the β-barrel domain of the B-subunit (Fig. 3a, b). The two EG stalks have the same structures in the current model, both adopting conformations that match the crystal structure[27], while their conformations in the complex are different when superimposing the β-barrel domain of the B-subunit (Fig. 3b). The two extensive stalks extend along the external surface of the B-A interface and reach down to the N-terminal domain of the a-subunit, where the density is not well resolved. It appears that these flexible sites may act as a hinge or elbow connecting the rigid stalks between the N-terminal domain of the E- and a-subunits. Similar structural flexibility of the peripheral stalk is observed in $F_oF_1$ structure where the rigid coiled-coil pillar is connected to $F_1$ via a flexible hinge[40]. Thus the stalk flexibility is likely to be a common feature of the rotary ATPases, allowing the rigid coiled-coil pillar to adjust position during a catalytic cycle.

The large data set of single particle images of the V/A-ATPase enabled identification of state3, missing in the previous study[9]. Indeed a comparison of the three rotational states reveals dynamic movements of each domain in the complex; the coiled-coil of the D-subunit relative to the d-subunit, the two peripheral stalks relative to the β-barrel domain of $A_3B_3$, and the hydrophilic N-terminal domain of the a-subunit. A clear dynamic rearrangement of the stator occurs during transition between each rotational state, as visualized in the Supplementary Movie 1–3. Similar movements have been reported in yeast V-ATPase, but motions of the V/A-ATPase appear different from those of yeast V-ATPase. Superimposition of the d-subunit in state1–3 of V/A-ATPase reveals that the orientation of the C-terminal helix of the D-subunit is very similar, but a large movement is observed in state3 relative to that of state1 and state2 (Supplementary Figure 7b). In contrast, movement of D-subunit helix is observed between each of the three rotational states in yeast V-ATPase (Supplementary Figure 7c). The circular motions of the two EG stators of V/A-ATPase is also apparently different from the sliding motion of the three EG stalks in yeast V-ATPase in the three rotational states (Supplementary Figure 8). Accompanying the circular motion of the two EG stalks, the a-NT region undergoes an up-down motion at the junction with the a-CT region. This shaking motion of the hydrophilic region of the a-

subunit of V/A-ATPase is modest compared to that of the yeast V-ATPase. The different motions of the TthV/A-ATPase and the yeast V-ATPase are most likely due to the structural differences between the two ATPases; in particular the number of peripheral stators. The $V_1$ region of TthV/A-ATPase is connected to the $V_o$ region by only two EG stalks, in contrast to the three EG stators observed in the yeast enzyme. In state3 of the V/A-ATPase, the two EG stalks slightly pull the $V_1$ region, resulting in a minor tilt of the coiled-coil of the D-subunit relative to the d-subunit (Supplementary Figure 9). This distortion is due to the asymmetric arrangement of the two EG stalks relative to the three-fold pseudo symmetry in $V_1$, and is likely contributing to the instability of state3 in the V/A-ATPase. The different motions of the EG stalks and the hydrophilic region of the a-subunit are also likely due to the different peripheral stalk structures of the two ATPases.

Several lines of evidence have suggested that rotary ATPases have elastic coupling between the catalytic and membrane embedded regions to facilitate torque transmission between ATP hydrolysis/synthesis and proton translocation. We previously demonstrated that the intact V/A-ATPase exhibited 12 dwell positions per revolution, corresponding to the 12-fold symmetry of the c ring when powering $V_1$ by ATP. This is despite the fact that the $V_1$ DF shaft undergoes 120° rotations for every ATP hydrolysis[13]. Presumably, ATP hydrolysis in $V_1$ provides the energy slope to accommodate the ~30° steps in the direction of rotation. It is likely that the flexible nature of the stator apparatus described here is involved in the elasticity of the protein allowing accommodation of these 30° movements.

The EM structure of membrane embedded region of the V/A-ATPase provided a higher resolution structure of $a_1c_{12}$ than previous intact rotary ATPase structures, enabled the main chains of the a-subunit and the $c_{12}$ ring fit well into the EM map (Figs. 5a and 6 and Supplementary Figure 6). The two highly tilted α-helices, a-TM7 and a-TM8, and the adjacent four outer α-helices of the $c_{12}$ ring are clearly identified in this model, even though the $c_{12}$ ring has a lower density than the other membrane regions. The highly conserved R563, equivalent to R735 in the yeast enzyme, is positioned in the central region of a-TM7, where the a-subunit is in close proximity to the outer helix of c-subunit (Fig. 5b). Although the resolution of the outer helices of the c-subunit are not sufficient for assignment of amino acid residues, homology mapping of the c-subunit suggests that E63 lies close to R563 in the a-subunit (Fig. 5b). The Arg-Glu pair is seen in the yeast $V_o$ structure, where the salt bridge was assigned at the interface between the a-subunit and the c”-subunit[18]. The EM map of state1 without the solvent mask provides several insights into the proton translocation pathway in $V_o$ despite the lower resolution of the EM map compared to that of yeast $V_o$. Aqueous cavities were more clearly visible in the micelle on both sides of the protein (Fig. 5c–e) than in previous reports[8], probably due to the rigid structure of LMNG micelle. The cytoplasmic cavity provides direct access to the hydrophilic residues on a-TM7, reaching the Arg-Glu pair between a-TM7 and the outer helix of c-subunit, suggesting that this cavity corresponds to the cytoplasmic half channel. The periplasmic cavity in the center of micelle shell faces the helix bundle composed of a-TM1, a-TM2, and a-TM3, including a line of hydrophilic residues, and connects to the aqueous hole giving an access to the inclined helices. In the EM structure of yeast $V_o$, the periplasmic proton entrance is not clear due to the presence of additional density from the hydrophilic loop region of the a-subunit. It is noteworthy that the yeast $V_o$ adopts an auto-inhibited form without proton channel activity[41]. Future work will focus on obtaining a higher resolution EM map of the TthV/A-ATPase to provide a more detailed view of both half channels.

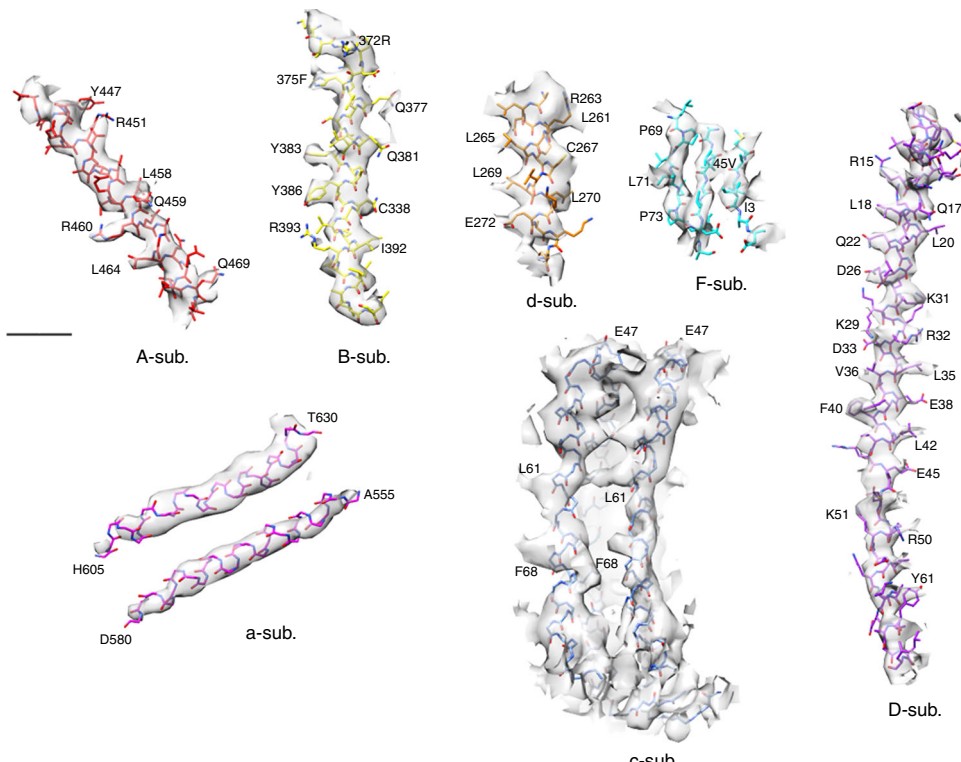

**Fig. 6** Examples of the fit of the model and density maps of state1. Amino acids for which side chain density was observed are indicated. Experimental maps are shown in semi-transparent gray. Scale bar, 10 Å

## Methods

**Preparation of *Tth*V/A-ATPase for Cryo-EM imaging**. The *Tth* V/A-ATPase containing His3 tags on the C-terminus of the c-subunit was isolated from *Thermus thermophilus* membranes[10] with some modifications. The membranes were prepared by disruption of aerobically cultured cells. After washing the membranes, the enzyme was solubilized from the membranes with 10% Triton X-100. Then, the enzyme was purified by $Ni^{2+}$-NTA affinity and a 6 ml Resource Q (Amersham) anion exchange column equilibrated with 0.03% dodecyl-β-D-maltoside (DDM). The purified enzyme was applied to a 1 ml of Resource Q column for detergent exchange to Lauryl Maltose Neopentyl Glycol (LMNG) by washing the column with a 20 mM Tris-HCl pH 8.0, 0.1 mM EDTA, pH 8.0 (TE buffer) containing 0.03% LMNG (Anatrace) for 105 min at a flow rate of 0.5 ml/min. The *Tth* V/A-ATPase was eluted by linear NaCl gradient using TE buffer (0–500 mM NaCl, 0.03% LMNG). The eluted fractions were concentrated and then passed through a gel-filtration column equilibrated with TE buffer containing 0.003% LMNG (Superdex 200; GE Healthcare). The peak fraction was collected and concentrated to ~ 0.03 mg/ml with TE buffer containing 100 mM NaCl.

**Cryo-EM imaging of *Tth*V/A-ATPase**. Sample vitrification was performed using a semi-automated vitrification device (Vitrobot, FEI). The 2.4 μl of sample solution at a concentration of 0.027 mg/ml was applied to glow discharged Quantifoil R2/2 with thin carbon backing (carbon-coated grids) in the Vitrobot at 100 % humidity, at 4 °C. The grid was then automatically blotted once from both sides with filter paper for 6 s blot time. The grid was then plunged into a liquid ethane with no delay time. Cryo-EM imaging was performed with Titan Krios (FEI, Eindhoven, Netherlands) operating at 300 kV acceleration voltage and equipped with a direct electron detector Falcon II (FEI, Eindhoven, Netherlands) in automated data collection mode at a calibrated magnification of 1.4 Å/pixel (×59,000) and dose of 26.4 $e^-$ $Å^{-2}$ (or 3.3 $e^-$ $Å^{-2}$ per frame) (where $e^-$ specifies electrons) with total 0.94 s exposure time. The data were collected as 7 movie frames excluding the first frame per an image (Supplementary Figure 2).

**Image processing**. A total of 4674 images were collected in 5 data sets, and which were combined. Whole-image drift was corrected and averaged using MOTION-CORR[42] for 1–7 movie frames of each image. Averaged images were used for determination of contrast transfer function (CTF) parameters with CTFFIND4[43] and selection of coordinates for particles images with RELION2.0 and 2.1b for all subsequent steps[44].

A subset of ~1000 particles was picked manually from a dataset, extracted using a 236 x 236 pixel box and subjected to 2D classification. At this point, a number of side views were selected (Supplementary Figure 2 and 10). Some of the resulting 2D class averages showed *Tth* V/A-ATPase details, which were low-pass filtered to 30

Å to these templates to limit model bias, and used as references for automatic particle picking of a data set. The automatically picked particles were screened manually to remove false positives, resulting particles were subjected to reference-free 2D classification. We selected particles from good 2D classes from a dataset, and then combined these particles for 3D classification (Supplementary Figure 2c).

223,982 particles selected from good 2D classes were used for 3D classification imposing a solvent mask of the whole complex to remove the noise in the solvent region followed by 3D classification into 12 classes (Supplementary Figure 3a). The resulting 12 classes were assessed manually and those representing similar states were merged. (Supplementary Figure 3a and b). Particles contributing to each state were refined using a solvent mask. At this point, state1 with 129,663 particles yielded a 5.1 Å, state2 with 33,587 particles yielded a 6.9 Å and state3 with 15,678 particles yielded a 8.3 Å (Supplementary Figure 3b).

Masks for 3D refinement were created by UCSF Chimera[28] and RELION program. PDB data file of whole complex (PDBID: 5GAR) was converted to a density map by molmap command of Chimera software applying 20–30 Å lowpass filter. Then, in RELION, the map was binarized and a soft edge was added.

**Focused classification and refinement with signal subtraction**. As previously observed, the two domains of the rotary motor work through a rotary mechanism, which involves structural flexibility[24]. Each state corresponded to a different position of the central rotor and thus introduced structural heterogeneity into $V_o$ domain, so we employed a focused 3D classification approach with signal subtraction to address this. The outline of image processing for each state is as follows. A soft mask of the $V_1EGd$ containing the micelle region of the whole complex was created by using molmap command of Chimera, relion_mask_create and relion_relion_image_handler program. The signals covered with the soft mask were subtracted from each particle in a state, and then a focused 3D classification was performed on the subtracted images that only have signal from $V_o-adc_{12}$ (Supplementary Figure 3c). Each state divided unequally after a focused 3D classification, and a major class in each state was refined on original images. The major class of state1 with 117,938 particles yielded a 5.0 Å, that of state2 with 30,802 particles yielded a 6.7 Å and that of state3 with 13,851 particles yielded a 7.5 Å overall resolution as a final EM map (Fig. 1, Supplementary Figure 3d). All resolutions are based on the gold-standard Fourier shell correlation (FSC) = 0.143 criterion (Supplementary Figure 3). These EM maps were deposited into EMDB as EMD-6810 for state1, EMD-6812 for state2, and EMD-6813 for state3.

**Masked 3D refinement approach with signal subtraction**. The solvent mask was constructed from the $V_1EGd$ region containing the $V_1$ domain, d-subunit, and C-terminal domains of the EG-subunits (E-subunit; E35-P146, G-subunit; E60-P120). A new dataset of experimental particle images that only have signal from the

$V_1EGd$ region were used for 3D refinement by the same subtraction method used previously (Supplementary Figure 3c). A focused 3D refinement of the subtracted images gave improved maps at a resolution of 4.7 Å in state1 (EMDB ID; EMD-6811) and 5.9 Å in state2, and did not give an improved map at a resolution of 8.1 Å in state3 (Supplementary Figure 3e).

**Model building and refinement**. The PDB accession numbers for atomic models used to interpret the EM maps were 3W3A ($A_3B_3DF$ complex[25]), 3GQB ($A_3B_3$ complex[24]), 3K5B (EG complex[27]), 3RRK (soluble part of a-subunit[38]) and 1R5Z (d-subunit[23]). The homology model for the a-subunit was calculated with Phyre2[45] using the V/A-type $H^+$-ATPase a-subunit from *T. thermophilus*[9] (PDB ID; 5GAR). A homology model for the c-subunit was generated with the corresponding subunit structures from *Saccharomyces serevisiae*[18] (PDB ID; 5TJ5) and *Enterococcus hirae*[46] (PDB ID; 2BL2) using MODELLER[47], built in UCSF Chimera program. Crystal structures and homology models were fitted into the three maps and their conformations refined by molecular dynamics flexible fitting[21] (MDFF) program. Due to the presence of a presumably flexible loop in a-subunit (residues 288–316) for which little density was present in the map, this region of sequence was truncated before MDFF. Atomic models calculated by MDFF were manually inspected for the map agreement by using COOT[48]. The fitting of the B-subunit coordination was improved by MDFF, thereby K5 and K6 structure assignments were corrected, by real space fitting to the density. Similarly, N-terminal structures of the E- and G-subunits were corrected by real space refinement to avoid steric clashing with the a-subunit, and some N- and C-terminal was removed from the crystal structure when the density was not clear. All figures were prepared using UCSF Chimera.

**Data availability**. Cryo-EM maps and the coordinates have been deposited in the Electron Microscopy Data Bank and Protein Data Bank with accession codes EMDB-6810 and PDB 5Y5X for state1, EMDB-6811 and PDB 5Y5Y for the peripheral domain of state1, EMDB-6812 and PDB 5Y5Z for state2 and EMDB-6813 and PDB 5Y60 for state3. The data supporting the findings of this study are available from the corresponding authors upon request.

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

## Acknowledgements

We thank Dr Bernadette Byrne for critical reading of our manuscript, and members of Yokoyama Lab for help and technical assistance. This work was supported by Grants-in-Aid from "Nanotechnology Platform" of the Ministry of Education, Culture, Sports,

Science and Technology (MEXT) to K.M. (Project No. 12024046) and Grants-in-Aid from the Ministry of Education, Culture, Sports, Science and Technology (MEXT) to K.Y. (Project No. 17H03648), and J.K. (Project No. 16K21472).

## Author contributions

A.N., J.K. designed, performed, and analyzed the experiments. A.N., J.K., K.M. analyzed the data and contributed to the preparation of the figures. M.T. constructed vectors for expression of mutant proteins. K.M. gave technical support and conceptual advice. K.Y. designed and supervised the experiments and wrote the manuscript. All authors discussed the results and commented on the manuscript.

## Additional information

**Competing interests:** The authors declare no competing financial interests.

