## [Peer Review File · Nature Communications]

Reviewer #1 (Remarks to the Author):

The writing of the manuscript needs significant improvement to make it clear to general audience. In addition to some detailed points listed below, which are meant to improve the readability of the manuscript, the only major point this reviewer would like addressed, is referred to a recently discovered bug issue with Relion, the software the authors used for their refinement. That issue could lead in some cases to overestimated resolution and in turn, to over-refinement of a structure. In any case, this should not change authors conclusions in any major way, but rather the final resolution.

Specific points:

373: In the light of recent Relion2 bug report (<https://www.jiscmail.ac.uk/cgi-bin/webadmin?A2=ind1708&L=CCPEM&F=&S=&P=40316>), in some cases the resolution was overestimated and thus there was a substantial over refinement. Author needs to verify that the version of Relion used is not affected by the bug. If so, re-calculation of all maps become necessary.

133-134: "This unequal population of the classes suggests that state1 of Tth V/A-ATPase represents the most energetically stable structure of the ATPase". Without duplication, this conclusion should be weakened.

141: It needs to be mentioned, that as those maps were improved in the V1EGd with this approach, the ac12 became less defined.

144: Please add more detail about the molecular dynamics: Was MDFF employed on the whole model, with ac12 being distorted part as well, or was it constrained?

167: "At the secondary structure level, there is also good agreement between the EM map and model in both state2 and state3, allowing differences in the conformations of the three different states to be detected". You built a model using homology modeling and X-ray structures, further using MDFF to fit these models into the density. It would be unlikely that there would be discrepancies, since you are using the EM density as a guideline. Do you rather mean that the quality of the EM density permitted construction of the model also in state2 and state3, and thus corresponding models showed conformational changes?

230: While the V1EGd region is well resolved, one has to address the fact, that the ac12 part in all the maps, from the 4.8 to the 7.5Å looks distorted. This means, that while focused classification (Supplementary Figure 2c) might have divided the dataset into three different subclasses, when refined, it uses the overall particle for alignment, which clearly favors the V1EGd region. Have authors tried doing focused, masked refinement of the ac12 part, where the V1EGd region is masked out and alignment could potentially allow them to resolve ac12 at a higher resolution? Figure 1a for state1 in this region looks distorted, Figure 4a clarifies this a bit, but shows helices are not always well resolved. Could authors provide their take on this?

381: "combined these particles from 5 datasets (Supplementary Fig. 1b)" – is it a reference to a wrong figure, or have you collected more datasets?

383: What do you mean by "specific mask"?

415: "This process created a new dataset" – which process? It is implied that there is a masking event of V1EGd region, but it is not stated before. If the form is to be kept, at 413: "containing" should be changed to "contains".

416: "images that only have signal from the subtracted images that only have signal from V1EGd region" – if I understand this correctly, you mean: "images that only have signal from V1EGd region"

419: "that is consistent with the higher resolution map than each state" – what do you mean?

427: "with the identical subunit structures" – Do you mean "related"? If it is identical, there is no need for homology model.

433: "assignment of the B subunit" – I believe that the position is known. Do you mean is "fit"?

435, 436: "real space fitting" – more commonly used expression is "real space refinement"

Figure 2a: It would be good to indicate the reason why this map of state1 is so different than in Figure 1a – by making it explicit that this is the map with the focus on the V1EGd region.

Supplementary Figure 1c: Are those class averages from the overall dataset (all particles), or from

one of the states (state1, state2, state3)? Please specify which one.

Supplementary Figure 4: Please state which of the states this is from.

Supplementary Figure 5: Please state whether these maps are the ones after focused refinement described in Supplementary Figure 2e or 2d. I assume they are from 2e.

Simple typo or language suggestion:

98: "the both whole" -> "both the whole"

105: Typo - Krios

99: "image subtraction masking method" - "signal subtraction", "masking" and "masked refinement/classification"

115: "signal subtraction masked method" - "signal subtraction", "masking" and "masked refinement/classification" or simply here "signal subtraction"

190: "by Resmap"

208: Typo: "model"

327: "In the EM map of state1 without solvent density subtraction, the LMNG micelle shell is still clearly visible" - I believe the term "solvent masked out during post-processing" is clearer

384: "Resulting 12 classes were classified based on their orientation of the F subunit by eyes, and the same classes were combined as a state" - This is a very convoluted way to say: "Resulting 12 classes were assessed manually and those representing similar states were merged"

386: "Particles contributing to each state were used for 3D refinement with mask to remove the noise in the solvent region followed by 3D refinement" - This is a very convoluted way to say: "Particles corresponding to each state were refined using a solvent mask"

392: "Then, the map was binarized and added softedge by Mask Creation job type on RELION" - I suggest using the following, to simplify "Then, in Relion, the map was binarized and a soft edge was added"

397: "Each state corresponding to different positions of central rotor seemed to have the structural heterogeneity of Vo domain, so we employed a focused 3D classification approach with signal subtraction to overcome the structural heterogeneity of Vo domain in each state" - I suggest simplifying : "Each state corresponding to different positions of central rotor exhibited structural heterogeneity of Vo domain, so we employed a focused 3D classification approach with signal subtraction to address this"

492: References have various formatting styles.

Reviewer #2 (Remarks to the Author):

The V/A type ATPase from *Thermus thermophilus* has been the model enzyme for the study of intact rotary ATPase structure for the last decade or so, with multiple reconstructions pushing the resolution barrier. Here Nakanishi and Kishikawa et al. present the highest resolution structure to date of this intact complex, revealing for the first time in this system three conformations related by 120 degree rotations of the central stalk.

The work in my opinion is scientifically sound, with high quality images, 2D classifications and density presented. The manuscript is well written, and at times provides new insight into this protein complex which is supported by the data shown. However, upon reading the manuscript I felt the literature had not been well referenced, and moreover claims of novelty and impact were overstated on multiple occasions throughout the manuscript. This manuscript really presents the

4.8 Å resolution reconstruction of a complex for which data to 6.4 Å for the entire complex and 4.5 Å for the V1 motor already exists. The authors appear to use language to subjectively ignore the previous data. In my opinion, this manuscript should describe what improvements have been made with this data and what new biology can be obtained from it, rather than brushing over this previous work, claiming new insight and comparing the data to other rotary ATPase subtypes to increase the impact of their findings.

A brief summary:

(i) the resolution of the complex is not drastically different to that seen in Schep et al. PNAS 2016. 113(12), 3245-50. It is higher, however the text reads as though vast improvements have been made without any real comparison to show this.

(ii) multiple conformations showing dynamic movements of rotary ATPases have been seen before (e.g. Zhao et al. Nature. 521(7551), 241-5 (2015))

(iii) the mechanism of proton translocation has been described previously in related systems to much greater detail (e.g. 3.9 Å Mazhab-Jafari et al. Nature 2016. 530(7590), 118-22)

(iv) the V1 mechanism presented has been described by a similar resolution crystal structure (Numoto et al. EMBO Rep. 10(11), 1228-34 (2009))

I would heavily recommend these be rectified before publication and more strength put towards the true impact of this work, namely State3 of this rotary catalytic mechanism and the dynamic interactions of this rotary ATPase subtype.

Please find below a list of revisions I feel necessary prior to publication:

(1) Line 34 – “These new maps are higher resolution than any obtained previously,...”

The Vo domain of the V type ATPase from *Saccharomyces cerevisiae* has been solved to 3.9 Å resolution (Mazhab-Jafari et al. Nature 2016. 530(7590), 118-22), which shares a common architecture and similar membrane sequence to that of *T. thermophilus* ATPase. I would suggest rephrasing this to “..higher resolution than any intact rotary ATPase obtained previously...” or “These new maps are higher resolution than any obtained previously on this complex,...”.

(2) Line 36 – “..as well as suggesting a clear proton translocation path...”

This is not the first time such a path has been suggested for a rotary ATPase, nor for a V/A type ATPase (see Kühlbrandt and Davies, Trends in Biochemical Sciences 2016. 41(1), 106-16 figure 6 and Mazhab-Jafari et al. Nature 2016. 530(7590), 118-22 figure 4 and Schep et al. PNAS 2016. 113(12), 3245-50 figure 5 for an examples). I would recommend changing this to “..as well as confirming the translocation path previously suggested by others...”.

(3) Line 124 – “Available crystal structures and homology models were docked...”

Given that multiple papers from the Rubinstein lab (e.g. PDBID; 5GAR) have used similar techniques on similar resolution maps, can the authors comment on what the differences are between these new models and that published previously? A nice supplementary comparison figure showing greater confidence in certain areas, or a figure describing the similarities or differences of the structures would go a long way to showing what new knowledge has been generated in this study.

(4) Line 126 – “State1 described here, adopted by the largest population of protein molecules, most closely corresponds to a previous cryo-EM map of *Saccharomyces cerevisiae* V-ATPase designated state3”

Why have the authors decided to compare this to the *S. cerevisiae* data, when similar studies have been done on this exact complex to similar resolution (6.4 Å - Schep et al. PNAS 2016. 113(12), 3245-50) showing 2 states? Can the authors comment on which states their states compare to? Are there any real differences between the conformations etc. using LMNG over DDM?

(5) Line 133 – “This unequal population of the classes suggests that state1 of *Tth* V/A-ATPase

represents the most energetically stable structure of the ATPase.”

This finding has been almost carbon copied from Schep et al. PNAS 2016. 113(12), 3245-50: “The uneven distribution of particle images between the two classes suggests that the highly populated class represents an energetically favorable ground state of the enzyme”. This work should at least be cited here, e.g. “..as has been previously seen in a similar study.”

(6) Line 145 – “The three AB pairs in A3B3 adopt different three conformations known as ‘open’, ‘closed’, and ‘semi-closed’, as seen in previously reported structures of the F1 and V1 complexes. In this model, strong positive density was observed near the P-loop in two AB pairs; the ‘closed’, and ‘semi-closed’ state.” And Lines 255-263 in the discussion.

What new information has been learnt in the study presented here? Numoto et al. EMBO Rep. 10(11), 1228-34 (2009) also presented an asymmetric structure with ADP bound in two of the Beta subunits. What is the RMSD between the final EM model and this crystal structure? Are the ADPs in the same position?

(7) Line 156 – “..has no ATPase activity.”

Have ATPase regeneration assays been performed on the LMNG solubilized protein? Detergents such as LDAO can have large effects on ATPase activity in other rotary ATPase systems.

(8) Lines 245-252 – The authors seem to claim that only the cytoplasmic pore has been previously reported. Many other groups have reported the equivalent periplasmic half channel before (e.g. Kühlbrandt and Davies, Trends in Biochemical Sciences 2016. 41(1), 106-16 figure 6). I would recommend rewriting this to include the work in light of others.

(9) Line 231 – “...resolution allowed us to build an atomic model of the whole complex.”

The data presented for the membrane region (figure 5a and supp. Fig. 4) is clearly not atomic resolution. I would suggest the authors remove this term “atomic” and just write model. It appears by the methods, that the authors used PDBID; 5GAR as a basis for their homology model. So again, my question would be; what new information has been learnt here? A good supplementary figure describing the docked model and what new information has been learnt would greatly improve this manuscript. For example, did Schep et al. dock the sequence incorrectly using the covariance method?

(10) Line 298 – “It appears that these flexible sites may act as a hinge or elbow connecting the rigid stalks between the N-terminal domain of the E and a-subunits.

Vinothkumar et al. PNAS 2016. pii: 201615902 see a similar hinge in the F-ATPase “...an elbow or joint allows the stator to bend to accommodate lateral movements during the activity of the catalytic domain”. It would be nice to include this work here and compare it to the presented data. Is the joint similar etc.?

Other minor comments:

(11) Line 76 – “..giving 120 degrees step each ATP hydrolysis...”

Many studies have shown the “1” motor to have a least one other substep in ATPase mode (e.g. Watanabe et al. Protein Sci. 2014 23(12):1773-9). It might be worth rephrasing this to remove confusion.

(12) Line 105 – “..Kraios (FEI)...”

Replace with Krios

(13) Line 166 – “..close interaction between subunit E/160-164 in both E subunit and subunit B/5-9,...”

Can the cross-linking data performed by Arata et al. J Biol Chem. 2002 Feb 1;277(5):3357-63 on the homologous V-ATPase be used to confirm the docking of the sequence in this area?

(14) Line 181 – “The linker region of the a-subunit connecting the hydrophilic and hydrophobic regions also shows weak density in the map of state1, suggesting that the regions connecting the stalk region of EG and a-NT are highly flexible”

I think it would be nice here to show this in a figure. Could the opposite of Figure 3b be included as supplementary, with subunit a being fixed and the stators moving?

(15) Line 273 – “The auto-inhibited form of V/A-ATPase presented here is advantageous as it hampers consumption of ATP when proton motive force is lost.”

This sentence is a little confusing, I would suggest rephrasing it.

Response to reviewer's comments:

Below we reproduce reviewer comments in black, present our responses in red, and the modified text in italic. The line number where modified text has been inserted is also provided.

Reviewer#1

1-1. Reviewers comment

The writing of the manuscript needs significant improvement to make it clear to general audience.

1-1. Our response:

According to the reviewer suggestion, we have rewritten the Introduction and included an additional figure, Supplementary Figure 1, to provide more information on the general background of the rotary ATPases. This is highlighted in red in the main manuscript.

1-2. Reviewers comment

In addition to some detailed points listed below, which are meant to improve the readability of the manuscript, the only major point this reviewer would like addressed, is referred to a recently discovered bug issue with Relion, the software the authors used for their refinement. That issue could lead in some cases to overestimated resolution and in turn, to over-refinement of a structure. In any case, this should not change authors conclusions in any major way, but rather the final resolution.

1-2. Our response:

We used RELION version 2.05, which contains the bug highlighted by the reviewer. We agree this may have caused an issue with our data and so we re-calculated all maps using RELION 2.1b. The final resolution for these new maps was also re-calculated followed by reconstruction of the models from the maps by MDFF. We have submitted these maps to the PDB. All figures have been re-drawn using the re-calculated results. These re-calculations do not affect in anyway our conclusions.

1-3. Reviewers comment

133-134: "This unequal population of the classes suggests that state1 of Tth V/A-ATPase represents the most energetically stable structure of the ATPase". Without duplication, this

conclusion should be weakened.

1-3. Our response:

To clarify the point, we rewrote this issue as follows;

126- The population of class 3 was less than ~7% of the total particles. This suggests that state3 of the TthV/A-ATPase represents the most energetically unstable structure of the ATPase. The molecular basis of the instability of state3 is discussed later.

We further cover the relative instability of state3 in the Discussion.

1-4. Reviewers comment

141: It needs to be mentioned, that as those maps were improved in the V1EGd with this approach, the ac12 became less defined.

1-4. Our response:

Accordingly we re-wrote the text as follows;

142- To overcome structural heterogeneity, we employed a masked refinement approach for the hydrophilic region. Each hydrophilic region was focused on, using a V1EGd soft mask to remove the affect of the remaining density, followed by refinement. The density of the V_o domain was blurred following this focused refinement.

1-5. Reviewers comment

144: Please add more detail about the molecular dynamics: Was MDFF employed on the whole model, with ac12 being distorted part as well, or was it constrained?

1-5. Our response:

For the molecular dynamics by MDFF, an almost complete model was used. As pointed out in the comment, no crystal structure is available for the membrane-spanning regions of the a- and c-subunits. However, a homology model produced from PDB Accession code 5GAR (ref. 9) fit our density for the membrane-spanning part of the a-subunit well. In addition, a homology model of the c-subunit produced from PDB Accession codes 5TJ5 (ref18) and 2BL2 (ref46) was a reasonable fit to our density. Thus, we clarify the use of the homology models by adding a

sentence,

“Combining the crystal structure of the hydrophilic region of the α -subunit (3RRK)³⁸ and the of the membrane embedded region, we could build an atomic model of almost the whole α -subunit within the overall complex.” In addition, the details of the molecular dynamics were added as *“For atomic model reconstruction, crystal structures²³⁻²⁷ and homology models were fitted into the three maps as rigid-bodies in Chimera ²⁸ and the conformations refined by MDFF with secondary structure restraints.”*

Together, we modified the text as follows;

147- For atomic model reconstruction, crystal structures²³⁻²⁷ and homology models were fitted into the three maps as rigid-bodies in Chimera ²⁸ and the conformations refined by MDFF with secondary structure restraints.

238- The improved map of state1 at 5.0 Å resolution allowed us to build a model of the whole complex. Combining the crystal structure of the hydrophilic region of the α -subunit (3RRK)³⁸ and the of the membrane embedded region, we could build an atomic model of almost the whole α -subunit within the overall complex. The main chain models of α -CT fitted well into the EM map, providing a more reliable model for the membrane domain of the whole complex than has previously been reported (Fig.5 and Supplementary Fig. 6).

1-6. Reviewers comment

167: “At the secondary structure level, there is also good agreement between the EM map and model in both state2 and state3, allowing differences in the conformations of the three different states to be detected”. You built a model using homology modeling and X-ray structures, further using MDFF to fit these models into the density. It would be unlikely that there would be discrepancies, since you are using the EM density as a guideline. Do you rather mean that the quality of the EM density permitted construction of the model also in state2 and state3, and thus corresponding models showed conformational changes?

1-6. Our response:

Thank you for correcting our statement properly. Our main statement is that the corresponding models of the EG stalks showed conformational changes. In addition, we would like to say that the fitted models do not show any severe distortions in their secondary structures. Thus, we

modified the text as follows;

174- While the fitting of the secondary structure elements to the EM maps was reasonable in both state2 and state3, corresponding models showed clear conformational changes of the EG stalk.

1-7. Reviewers comment

230: While the V1EGd region is well resolved, one has to address the fact, that the ac12 part in all the maps, from the 4.8 to the 7.5Å looks distorted. This means, that while focused classification (Supplementary Figure 2c) might have divided the dataset into three different subclasses, when refined, it uses the overall particle for alignment, which clearly favors the V1EGd region. Have authors tried doing focused, masked refinement of the ac12 part, where the V1EGd region is masked out and alignment could potentially allow them to resolve ac12 at a higher resolution? Figure 1a for state1 in this region looks distorted, Figure 4a clarifies this a bit, but shows helices are not always well resolved. Could authors provide their take on this?

1-7. Our response:

As the reviewer mentioned, we firstly tried signal subtraction for V1EGd to classify the ac12 region. Unfortunately, no meaningful 3D class average was obtained, probably due to the small size of the ac12 region. In our structure of V/A-ATPase, the LMNG micelle constitutes of rigid detergent shell, preventing a clear view of the membrane embedded domain. Most transmembrane helices are clearly visible when removing the density corresponding to the detergent micelle. Please see Supplemental figure 10, where EM density for the c12 ring is visible without a terrible distortion, when the density corresponding to the detergent micelle has been removed.

1-8. Reviewers comment

381: “combined these particles from 5 datasets (Supplementary Fig. 1b)” – is it a reference to a wrong figure, or have you collected more datasets?

1-8. Our response:

As the reviewer has pointed out, the wrong figure has been cited here. We have modified the text as follows;

428- “combined these particles for 3D classification (Supplementary Fig. 2c)”

1-9. Reviewers comment

383: What do you mean by “specific mask”?

1-9. Our response:

This refers to the whole complex mask. To clarify this point, we have modified the text as follows 430- “solvent mask of the whole complex.”

1-10. Reviewers comment

415: “This process created a new dataset” – which process? It is implied that there is a masking event of V1EGd region, but it is not stated before. If the form is to be kept, at 413: “containing” should be changed to “contains”.

416: “images that only have signal from the subtracted images that only have signal from V1EGd region” – if I understand this correctly, you mean: “images that only have signal from V1EGd region”

1-10. Our response:

As the reviewer mentioned, this description is not clear. We have modified the text as follows:

457- The solvent mask was constructed from the V₁EGd region containing the V₁ domain, d subunit, and C-terminal domains of the EG-subunits (E-subunit; E35-P146, G-subunit; E60-P120). A new dataset of experimental particle images that only have signal from the V₁EGd region were used for 3D refinement by the same subtraction method used previously (Supplementary Fig. 3c).

1-11. Reviewers comment

419: “that is consistent with the higher resolution map than each state” – what do you mean?

1-11. Our response:

As the reviewer mentioned, this sentence is unclear. Thus we removed the part.

461- A focused 3D refinement of the subtracted images gave improved maps at a resolution of 4.7 Å in state1 (EMDB ID; EMD-6811) and 5.9 Å in state2, and did not give an improved map of state3 (Supplementary Fig. 3e).

1-12. Reviewers comment

427: “with the identical subunit structures” –Do you mean “related”? If it is identical, there is no need for homology model.

1-12. Our response:

As the reviewer pointed out, this sentence is not correct. We modified it as follows;

469- “with the corresponding subunit structures”

1-13. Reviewers comment

433: “assignment of the B subunit” – I believe that the position is known. Do you mean is “fit”?

1-13. Our response:

As the reviewer mentioned, we changed the sentence as follows;

476- The fitting of the B-subunit coordination was improved by MDFF,

1-14. Reviewers comment

435, 436: “real space fitting” - more commonly used expression is “real space refinement”

1-14. Our response:

Thank you for this suggestion. We have modified the sentence as follows;

478- Similarly, N-terminal structures of the E- and G-subunits were corrected by real space refinement to avoid steric clashing with the a-subunit,

1-15. Reviewers comment

Figure 2a: It would be good to indicate the reason why this map of state1 is so different than in Figure 1a – by making it explicit that this is the map with the focus on the V1EGd region.

1-15. Our response:

According to the reviewer’s suggestion, we modified the figure 2 legend as follows;

505- a The EM map in state1 with focus on the V₁EGd (A₃B₃DF(EG_{CT})₂d) subdomain. The density of the membrane embedded region was subtracted by focused refinement using the mask of V₁EGd.

1-16. Reviewers comment

Supplementary Figure 1c: Are those class averages from the overall dataset (all particles), or from one of the states (state1, state2, state3)? Please specify which one.

1-16. Our response:

Those 2D class averages are from all particles. According to the reviewer suggestion, we modified the legend as follows;

Supplementary figure 2. legend- Example of 2D class averaged images from all particles obtained from reference-free classification.

1-17. Reviewers comment

Supplementary Figure 4: Please state which of the states this is from.

1-17. Our response:

Accordingly, we modified the legend title as follows;

Supplementary figure 10. Legend. Examples of the fit of the model and density maps of state1.

1-18. Reviewers comment

Supplementary Figure 5: Please state whether these maps are the ones after focused refinement described in Supplementary Figure 2e or 2d. I assume they are from 2e.

1-18. Our response:

Thank you for your suggestion. We modified the legend title as follows;

Supplementary figure 5. legend- Experimental maps focused on the V₁EGd region (Supplementary Fig. 3e) are shown in semi-transparent grey.

1-19. Reviewers comment

Simple typo or language suggestion:

1-20. Our response:

Thank you for your suggestions. Accordingly, we corrected the manuscript. The corrected parts in the text are highlighted in red.

Reviewer#2

2-1. Reviewers comment

The work in my opinion is scientifically sound, with high quality images, 2D classifications and density presented. The manuscript is well written, and at times provides new insight into this protein complex which is supported by the data shown. However, upon reading the manuscript I felt the literature had not been well referenced, and moreover claims of novelty and impact were overstated on multiple occasions throughout the manuscript. This manuscript really presents the 4.8 Å resolution reconstruction of a complex for which data to 6.4 Å for the entire complex and 4.5 Å for the V1 motor already exists. The authors appear to use language to subjectively ignore the previous data. In my opinion, this manuscript should describe what improvements have been made with this data and what new biology can be obtained from it, rather than brushing over this previous work, claiming new insight and comparing the data to other rotary ATPase subtypes to increase the impact of their findings.

A brief summary:

- (i) the resolution of the complex is not drastically different to that seen in Schep et al. PNAS 2016. 113(12), 3245-50. It is higher, however the text reads as though vast improvements have been made without any real comparison to show this.
- (ii) multiple conformations showing dynamic movements of rotary ATPases have been seen before (e.g. Zhao et al. Nature. 521(7551), 241-5 (2015))
- (iii) the mechanism of proton translocation has been described previously in related systems to much greater detail (e.g. 3.9 Å Mazhab-Jafari et al. Nature 2016. 530(7590), 118-22)
- (iv) the V1 mechanism presented has been described by a similar resolution crystal structure (Numoto et al. EMBO Rep. 10(11), 1228-34 (2009))

I would heavily recommend these be rectified before publication and more strength put towards the true impact of this work, namely State3 of this rotary catalytic mechanism and the dynamic interactions of this rotary ATPase subtype.

2-1 Our response;

We appreciate the suggestions of Reviewer 2. We agree that these will greatly enhance the clarity and context of our important results. We have completely rewritten the text in places as detailed below in order to address these issues;

2-1-1 reviewer's suggestion

the resolution of the complex is not drastically different to that seen in Schep et al. PNAS 2016. 113(12), 3245-50. It is higher, however the text reads as though vast improvements have been made without any real comparison to show this.

2-1-1 our response

We do believe our EM map of state1 at 5.0 Å resolution a dramatic improvement from the previous reported map at 6.4 Å resolution. In order to underline the precise nature of these improvements we have added Supplementary Fig. 6 which makes it much easier for readers to appreciate the difference between two maps. Also we modified the text as follows;

169- Our improved EM map provides more detailed structural information than the previously reported map (see Supplementary Fig. 6). Our higher resolution model of state1 reveals that there are close interactions between E/160-164 of E-subunits and B/5-9 of B-subunit, where rigid β -sheet structures are formed (Fig. 3a).

2-1-2 reviewer's suggestion

(ii) multiple conformations showing dynamic movements of rotary ATPases have been seen before (e.g. Zhao et al. Nature. 521(7551), 241-5 (2015)).

I would heavily recommend these be rectified before publication and more strength put towards the true impact of this work, namely State3 of this rotary catalytic mechanism and the dynamic interactions of this rotary ATPase subtype.

2-1-2 our response

This reviewer's suggestion is very important. Thus we carefully compared the three rotational states of the V/A-ATPase to those of the yeast V-ATPase, and found some differences with respect to the peripheral stalk structure. We discuss this point in the Discussion, as follows;

326- The large data set of single particle images of the V/A-ATPase enabled identification of state3, missing in the previous study⁹. Indeed a comparison of the three rotational states reveals dynamic movements of each domain in the complex; the coiled-coil of the D-subunit relative to

the d-subunit, the two peripheral stalks relative to the α -barrel domain of A_3B_3 , and the hydrophilic N-terminal domain of the a-subunit. A clear dynamic rearrangement of the stator occurs during transition between each rotational state, as visualized in the supplemental movie S1-3. Similar movements have been reported in yeast V-ATPase, but motions of the V/A-ATPase appear different from those of yeast V-ATPase. Superimposition of the d-subunit in state1-3 of V/A-ATPase reveals that the orientation of the C-terminal helix of the D-subunit is very similar, but a large movement is observed in state3 relative to that of state1 and state2 (Supplementary Fig. 7b). In contrast, movement of orientation of D-subunit helix is observed between each of the three rotational states in yeast V-ATPase (Supplementary Fig. 7c). The circular motions of the two EG stators of V/A-ATPase is also apparently different from the sliding motion of the three EG stalks in yeast V-ATPase in the three rotational states (Supplementary Fig. 8). Accompanying the circular motion of the two EG stalks, the a-NT region undergoes an up-down motion at the junction with the a-CT region. This shaking motion of the hydrophilic region of the a-subunit of V/A-ATPase is modest compared to that of the yeast V-ATPase. The different motions of the TthV/A-ATPase and the yeast V-ATPase are most likely due to the structural differences between the two ATPases; in particular the number of peripheral stators. The V_1 region of TthV/A-ATPase is connected to the V_o region by only two EG stalks, in contrast to the three EG stators observed in the yeast enzyme. In state3 of the V/A-ATPase, the two EG stalks slightly pull the V_1 region, resulting in a minor tilt of the coiled-coil of the D-subunit relative to the d-subunit (Supplementary Fig. 9). This distortion is due to the asymmetric arrangement of the two EG stalks relative to the three-fold pseudo symmetry in V_1 , and is likely contributing to the instability of state3 in the V/A-ATPase. The different motions of the EG stalks and the hydrophilic region of the a-subunit are also likely due to the different peripheral stalk structures of the two ATPases.

2-1-3 reviewer's suggestion

(iii) the mechanism of proton translocation has been described previously in related systems to much greater detail (e.g. 3.9 Å Mazhab-Jafari et al. Nature 2016. 530(7590), 118-22)

2-1-3 Our response

Although the resolution of our EM map is lower than that available for the yeast V_o , it has provided a wealth of information on the proton translocation pathway. We have rewritten the discussion to clarify this point.

373- The EM map of state1 without the solvent mask during post-processing provides several insights into the proton translocation pathway in V_o despite the lower resolution of the EM map

compared to that of yeast V_o . Aqueous cavities were more clearly visible in the micelle on both sides of the protein (Fig.5c-e) than previous reports⁸, probably due to the rigid structure of LMNG micelle. The cytoplasmic cavity provides direct access to the hydrophilic residues on α -TM7, reaching the Arg-Glu pair between α -TM7 and the outer helix of c -subunit, suggesting that this cavity corresponds to the cytoplasmic half channel. The periplasmic cavity in the center of micelle shell faces the helix bundle composed of α -TM1, α -TM2, and α -TM3, including a line of hydrophilic residues, and connects to the aqueous hole giving an access to the inclined helices. In the EM structure of yeast V_o , the periplasmic proton entrance is not clear due to the presence of additional density from the hydrophilic loop region of the a -subunit. It is noteworthy that the yeast V_o adopts an auto-inhibited form without proton channel activity⁴⁰. Future work will focus on obtaining a higher resolution EM map of the TthV/A-ATPase to provide a more detailed view of both half channels.

2-1-4 reviewer's suggestion

(iv) the V1 mechanism presented has been described by a similar resolution crystal structure (Numoto et al. EMBO Rep. 10(11), 1228-34 (2009))

2-1-4 Our response

One of advantage of single particle analysis using EM is to determine a native frozen state of enzyme, not affected by crystallization conditions. We have rewritten a section in the discussion to underscore the importance of this difference in the structure methods;

270- The V/A-ATPase was frozen on the grid in the absence of additional nucleotide, thus the bound ADP has co-purified with the protein directly from the cell either in the form of ADP or is the result of ATP hydrolysis. This is the first report of nucleotides bound to the rotary ATPase in the native frozen state. The previously reported crystal structure of TthV₁ at 3.9 Å resolution, also shows nucleotides occupying the two nucleotide binding sites, as a result of co-crystallisation with ADP. Thus, the V₁ region of our cryo-EM map seems to represent the same catalytic state as the crystal structure of V₁. The crystal structure of V₁ fits well into the cryo-EM density map at the secondary structure level, but the contact region of V₁ to both EG and d-subunit in the EM map differs slightly (Fig. 3 and 4).

2-2 Reviewer's comment

Line 34 – “These new maps are higher resolution than any obtained previously,…”

The V_o domain of the V type ATPase from *Saccharomyces cerevisiae* has been solved to 3.9 Å resolution (Mazhab-Jafari et al. Nature 2016. 530(7590), 118-22), which shares a common architecture and similar membrane sequence to that of *T. thermophilus* ATPase. I would suggest rephrasing this to “..higher resolution than any intact rotary ATPase obtained previously…” or “These new maps are higher resolution than any obtained previously on this complex,…”.

2-2 Our response

According to the reviewer’s suggestions, we rewrote the abstract to incorporate this suggestion. Also we carefully re-wrote sections of both the results and discussion to also reflect this. Please read our response to 2-1-3 above.

Abstract

33- These new maps are higher resolution than any intact rotary ATPase obtained previously,

Result

259- On the periplasmic side, another cavity is more clearly observed in the detergent micelle than previously reported for other EM maps of intact rotary ATPases^{8, 9,14-17}.

Discussion

373- The EM map of state I without the solvent mask during post-processing provides several insights into the proton translocation pathway in V_o despite the lower resolution of the EM map compared to that of yeast V_o .

2-3 Reviewer’s comment

Line 36 – “..as well as suggesting a clear proton translocation path…”

This is not the first time such a path has been suggested for a rotary ATPase, nor for a V/A type ATPase (see Kühlbrandt and Davies, Trends in Biochemical Sciences 2016. 41(1), 106-16 figure 6 and Mazhab-Jafari et al. Nature 2016. 530(7590), 118-22 figure 4 and Schep et al. PNAS 2016. 113(12), 3245-50 figure 5 for an examples). I would recommend changing this to “..as well as confirming the translocation path previously suggested by others…”.

2-3 Our response

Please read our response to 2-1-3 above.

2-4 Reviewer's comment

Line 124 – “Available crystal structures and homology models were docked...”

Given that multiple papers from the Rubinstein lab (e.g. PDBID; 5GAR) have used similar techniques on similar resolution maps, can the authors comment on what the differences are between these new models and that published previously? A nice supplementary comparison figure showing greater confidence in certain areas, or a figure describing the similarities or differences of the structures would go a long way to showing what new knowledge has been generated in this study.

2-4 Our response

Thank you for your kind suggestion. Please read our response to 2-1-1 above.

2-5 Reviewer's comment

Line 126 – “State1 described here, adopted by the largest population of protein molecules, most closely corresponds to a previous cryo-EM map of *Saccharomyces cerevisiae* V-ATPase designated state3”

Why have the authors decided to compare this to the *S. cerevisiae* data, when similar studies have been done on this exact complex to similar resolution (6.4 Å - Schep et al. PNAS 2016. 113(12), 3245-50) showing 2 states? Can the authors comment on which states their states compare to? Are there any real differences between the conformations etc. using LMNG over DDM?

2-5 Our response

This sentence was originally included to describe the orientation of the F subunit in state1 of the V/A-ATPase compared to state3 of the yeast V-ATPase. As the reviewer mentioned, the description is not necessary in this manuscript. Thus we removed the sentence in the revision.

2-6 Reviewer's comment

Line 133 – “This unequal population of the classes suggests that state1 of Tth V/A-ATPase represents the most energetically stable structure of the ATPase.”

This finding has been almost carbon copied from Schep et al. PNAS 2016. 113(12), 3245-50: “The uneven distribution of particle images between the two classes suggests that the highly

populated class represents an energetically favorable ground state of the enzyme”. This work should at least be cited here, e.g. “..as has been previously seen in a similar study.”

2-6 Our response

According to the reviewer suggestion, we focused on the structure of state3 in the revision. Thus, we modified the sentence as follows;

124- The V/A-ATPase classes corresponding to the different rotational states were populated more unequally than that seen in EM studies of the yeast V-ATPase or bovine F type ATPase¹⁵⁻¹⁷, and as seen in a similar study by Schep et al⁹. The population of the class3 was less than ~7% of the total particles. This suggests that state3 of the TthV/A-ATPase represents the most energetically unstable structure of the ATPase. The molecular basis of this instability of state3 is discussed later.

2-7 Reviewer’s comment

Line 145 – “The three AB pairs in A3B3 adopt different three conformations known as ‘open’, ‘closed’, and ‘semi-closed’, as seen in previously reported structures of the F1 and V1 complexes. In this model, strong positive density was observed near the P-loop in two AB pairs; the ‘closed’, and ‘semi-closed’ state.” And Lines 255-263 in the discussion.

What new information has been learnt in the study presented here? Numoto et al. EMBO Rep. 10(11), 1228-34 (2009) also presented an asymmetric structure with ADP bound in two of the Beta subunits. What is the RMSD between the final EM model and this crystal structure? Are the ADPs in the same position?

2-7 Our response

We are convinced that the advantage of single particle analysis using cryo-EM is determination of a native frozen state structure of the protein. Crystal structures of proteins can be affected by the crystallization conditions. For example, crystals of TthV₁ were grown in the presence of ADP, resulting in determination of the structure of the ADP bound form. From this study, we can surmise that the V/A-ATPase is likely to adopt the 2-ADP form in biological membranes. This is clearly an important consideration in understanding V/A-ATPase function. Our study reveals the first EM structure of a rotary ATPases with nucleotides bound in the catalytic sites. At the present resolution of the maps, the RMSD between the crystal structure and our model is

not meaningful. Therefore, we did not a detailed comparison of the V1 regions of the two structures but modified the text to clarify these points as follows;

270- The V/A-ATPase was frozen on the grid in the absence of additional nucleotide, thus the bound ADP has co-purified with the protein directly from the cell either in the form of ADP or is the result of ATP hydrolysis. This is the first report of nucleotides bound to the rotary ATPase in the native frozen state. The previously reported crystal structure of Tth V₁ at 3.9 Å resolution, also shows nucleotides occupying the two nucleotide binding sites, as a result of co-crystallisation with ADP. Thus, the V₁ region of our cryo-EM map seems to represent the same catalytic state as the crystal structure of V₁. The crystal structure of V₁ fits well into the cryo-EM density map at the secondary structure level, but the contact region of V₁ to both EG and d-subunit in the EM map differs slightly (Fig. 3 and 4).

2-8 Reviewer's comment

Line 156 – “..has no ATPase activity.”

Have ATPase regeneration assays been performed on the LMNG solubilized protein? Detergents such as LDAO can have large effects on ATPase activity in other rotary ATPase systems.

2-8 Our response

Unlike the effects seen for F₁-ATPase from bacteria, LDAO does not increase the ATPase activity of the V/A-ATPase. LMNG also has no affect on the activity.

2-9 Reviewer's comment

Lines 245-252 – The authors seem to claim that only the cytoplasmic pore has been previously reported. Many other groups have reported the equivalent periplasmic half channel before (e.g. Kühlbrandt and Davies, Trends in Biochemical Sciences 2016. 41(1), 106-16 figure 6). I would recommend rewriting this to include the work in light of others.

2-9 Our response

According to the reviewer's suggestion, we have rewritten text in two places to take this point into account as follows;

Introduction

90- The recent high resolution EM structure of yeast V_o provided a near atomic resolution model of the membrane embedded region responsible for proton translocation. This structure

revealed an apparent aqueous cavity accessible from the cytoplasmic side and capable of proton translocation, although no periplasmic proton pathway could be clearly discerned in this structure. A periplasmic channel had been suggested in some earlier low resolution EM maps of rotary ATPases⁸.

Discussion part

373- The EM map of state I without the solvent mask during post-processing provides several insights into the proton translocation pathway in V_o despite the lower resolution of the EM map compared to that of yeast V_o . Aqueous cavities were more clearly visible in the micelle on both sides of the protein (Fig.5c-e) than previous reports⁸, probably due to the rigid structure of LMNG micelle.

2-10 Reviewer's comment

Line 231 – "...resolution allowed us to build an atomic model of the whole complex."

The data presented for the membrane region (figure 5a and supp. Fig. 4) is clearly not atomic resolution. I would suggest the authors remove this term "atomic" and just write model. It appears by the methods, that the authors used PDBID; 5GAR as a basis for their homology model. So again, my question would be; what new information has been learnt here? A good supplementary figure describing the docked model and what new information has been learnt would greatly improve this manuscript. For example, did Schep et al. dock the sequence incorrectly using the covariance method?

2-10 Our response

We thank the reviewer for this good suggestion. Accordingly, we have removed the word "atomic" from this sentence. The geometry and modeling accuracy of 5GAR is not excellent overall. This is especially the case for the membrane embedded region. It is possible to see this very clearly using UCSF Chimera. Our improved EM map has allowed the production of a much more accurate model of the V/A-ATPase. To emphasize these points, we carefully rewrote the text and added a Supplementary figure comparing the 5GAR map with our new map (Supplementary fig. 6).

242- The main chain models of a-CT fitted well into the EM map, providing a more reliable model for the membrane domain of the whole complex than has previously been reported (Fig.5 and Supplementary Fig. 6).

2-11 Reviewer's comment

Line 298 – “It appears that these flexible sites may act as a hinge or elbow connecting the rigid stalks between the N-terminal domain of the E and a-subunits.

Vinothkumar et al. PNAS 2016. pii: 201615902 see a similar hinge in the F-ATPase “...an elbow or joint allows the stator to bend to accommodate lateral movements during the activity of the catalytic domain”. It would be nice to include this work here and compare it to the presented data. Is the joint similar etc.?

2-11 Our response

Thank you for this suggestion. We compared flexibility of the peripheral stalk of F_0F_1 and the V/A-ATPase and added the following text describing this in the discussion.

321- Similar structural flexibility of the peripheral stalk is observed in F_0F_1 structure where the rigid coiled-coil pillar is connected to F_1 via a flexible hinge⁴⁰. Thus the stalk flexibility is likely to be a common feature of the rotary ATPases, allowing the rigid coiled-coil pillar to adjust position during a catalytic cycle.

2-11 Reviewer's comment

Line 76 – “..giving 120 degrees step each ATP hydrolysis...”

Many studies have shown the “1” motor to have a least one other substep in ATPase mode (e.g. Watanabe et al. Protein Sci. 2014 23(12):1773-9). It might be worth rephrasing this to remove confusion.

2-11 Our response

In contrast to F_1 -ATPase, V_1 -ATPase does not show an additional substep in ATPase mode. In order to clarify this point, we added Supplementary Fig. 1 which illustrates this point.

2-12 Reviewer's comment

Line 105 – “..Kraios (FEI)...”

Replace with Krios

2-12 Our response

We replaced it.

2-13 Reviewer's comment

Line 166 – “..close interaction between subunit E/160-164 in both E subunit and subunit B/5-9,…”

Can the cross-linking data performed by Arata et al. J Biol Chem. 2002 Feb 1;277(5):3357-63 on the homologous V-ATPase be used to confirm the docking of the sequence in this area?

2-13 Our response

Arata et al showed that B/15A and B/45E are in close proximity to E subunit in the yeast V-ATPase using cross linking. However this is unlikely to be directly relevant to our structure since residues B/5-9 of V/A ATPase, which forming a close interaction with E160-164, in the V/A-ATPase corresponds to a different region of B, B/25-29, of the yeast V-ATPase. Please see the additional figure at page #21.

2-14 Reviewer's comment

Line 181- “The linker region of the a-subunit connecting the hydrophilic and hydrophobic regions also shows weak density in the map of state1, suggesting that the regions connecting the stalk region of EG and a-NT are highly flexible”

I think it would be nice here to show this in a figure. Could the opposite of Figure 3b be included as supplementary, with subunit a being fixed and the stators moving?

2-14 Our response

Thank you for this excellent suggestion. We also supplemented movieS1 that is showing the movement of the complex when TM domain of a-subunit is fixed, and that is added as a reference at Line 190.

2-15 Reviewer's comment

Line 273- “The auto-inhibited form of V/A-ATPase presented here is advantageous as it hampers consumption of ATP when proton motive force is lost.”

This sentence is a little confusing, I would suggest rephrasing it.

2-15 Our response

Accordingly, we modified the sentence as follows;

293- The ADP inhibited form of V/A-ATPase is physiologically advantageous for cells as it prevents consumption of ATP when proton motive force is lost.

Supplementary Fig. The sequence comparison of B subunit. The cross linked residues are indicated by arrows.

Reviewer #1 (Remarks to the Author):

This reviewer has no major issues with the manuscript. All the points raised in the previous round of review have been satisfactorily addressed. The authors reproduced their work to verify their results as requested by the reviewer. This was done in order to verify whether an existing bug in the software used for cryo-EM analysis hadn't changed the outcome of the experiment. The results were found to be reproducible and not overestimated.

The following issues were found to be very minor:

30: "bacterium, Thermus" – bacterium Thermus

144: Do the authors mean that there was a refinement inside a mask? If so, this sentence should be simplified, since it implies that first, there is focus, and then there is refinement, while both things happen during the same process. Since the previous sentence states "we employed a masked refinement approach for the hydrophilic region", this reviewer suggests the following change: "we employed masked refinement with V1EGd soft mask for the hydrophilic region" or "for each hydrophilic region". The sentence "Each hydrophilic region was focused on, using a V1EGd soft mask to remove the affect of the remaining density, followed by refinement" could then be removed to avoid confusion.

147: "atomic model reconstruction": "atomic model construction" is clearer since the atomic model is constructed once

197: "in the map are relatively high resolution": "in the map are at relatively high resolution"

198: "Resmaps": "Resmap"

240: "the of the": "of the"

336: "movement of orientation of D-subunit helix": "movement of D-subunit helix" or "change of orientation of D-subunit helix"

373: "without the solvent mask" suffices

376: "than previous reports": "than in previous reports"

463: "did not gave": "did not give" or "yield"

483: "maps and its coordinates": "maps and the coordinates"

Reviewer #2 (Remarks to the Author):

Overall I believe the manuscript to be improved, particularly with respect to the cited literature, figures to show the improvement in structural details and emphasis on the novel aspects of this work.

I therefore I would recommend the publication of this work, subject to the following small changes of the newly added text:

(i) lines 190-193, 317-318, 322-325 and 337-341.

The large section added to the discussion on the truly novel aspect of this work, namely "state 3" and the dynamic movements of the V/A ATPase, is a good addition to the manuscript. However, the "circular motion" (line 340) has been observed and predicted before as a "wobbling" of the A1 domain relative to the Ao domain by Stewart et al. (The dynamic stator stalk of rotary ATPases. Nature Communications. 2012. doi: 10.1038/ncomms1693). In this work the dynamic nature of the EG stalk regions was also commented on. Are the movements Nakanishi et. al observe similar or different? The text seems a little confusing, in that the stalks are "rigid and inflexible" (line 318), but "bend" at the same time (line 191). I feel this point should be clarified and discussed.

(ii) lines 273-274. "This is the first report of nucleotides bound to the rotary ATPase in the native frozen state." and 2-7 in the response "Our study reveals the first EM structure of a rotary ATPases with nucleotides bound in the catalytic sites."

This statement isn't correct, Sobti et al. 2016 (reference 17) clearly identify nucleotides in their map of E. coli ATP synthase (see Figure 5). The text should be edited accordingly.

Response to reviewer's comments:

Below we reproduce reviewer comments in black, present our responses in red, and the modified text in italic. The line number where modified text has been inserted is also provided.

Reviewer#1

1-1. Reviewers comment

This reviewer has no major issues with the manuscript. All the points raised in the previous round of review have been satisfactorily addressed. The authors reproduced their work to verify their results as requested by the reviewer. This was done in order to verify whether an existing bug in the software used for cryo-EM analysis hadn't changed the outcome of the experiment. The results were found to be reproducible and not overestimated.

The following issues were found to be very minor:

30: "bacterium, Thermus" – bacterium Thermus

144: Do the authors mean that there was a refinement inside a mask? If so, this sentence should be simplified, since it implies that first, there is focus, and then there is refinement, while both things happen during the same process. Since the previous sentence states "we employed a masked refinement approach for the hydrophilic region", this reviewer suggests the following change: "we employed masked refinement with VIEGd soft mask for the hydrophilic region" or "for each hydrophilic region". The sentence "Each hydrophilic region was focused on, using a VIEGd soft mask to remove the affect of the remaining density, followed by refinement" could then be removed to avoid confusion.

147: "atomic model reconstruction": "atomic model construction" is clearer since the atomic model is constructed once

197: "in the map are relatively high resolution": "in the map are at relatively high resolution"

198: "Resmaps": "Resmap"

240: "the of the": "of the"

336: "movement of orientation of D-subunit helix": "movement of D-subunit helix" or "change of orientation of D-subunit helix"

373: "without the solvent mask" suffices

376: "than previous reports": "than in previous reports"

463: "did not gave": "did not give" or "yield"

483: "maps and its coordinates": "maps and the coordinates"

1-1. Our response:

We thank for the referee's suggestions. Accordingly, we corrected the manuscript. The corrected parts in the text are highlighted in red.

Reviewer#2

2. Reviewers comment

Overall I believe the manuscript to be improved, particularly with respect to the cited literature, figures to show the improvement in structural details and emphasis on the novel aspects of this work. I therefore I would recommend the publication of this work, subject to the following small changes of the newly added text:

2. Our response;

We would like to thank the reviewer for meaningful suggestions. We agree that these will greatly improve the clarity and context of our important results. We have considered the suggestions carefully, and offered a response to these as below in order to address these issues;

2-1 reviewer's suggestion

(i) lines 190-193, 317-318, 322-325 and 337-341.

The large section added to the discussion on the truly novel aspect of this work, namely "state 3" and the dynamic movements of the V/A ATPase, is a good addition to the manuscript. However, the "circular motion" (line 340) has been observed and predicted before as a "wobbling" of the A1 domain relative to the Ao domain by Stewart et al. (The dynamic stator stalk of rotary ATPases. Nature Communications. 2012. doi: 10.1038/ncomms1693). In this work the dynamic nature of the EG stalk regions was also commented on. Are the movements Nakanishi et. al observe similar or different?

2-1 Our response

These are different. The predicted movement of EG by Stewart et al was dramatically dynamic. In their simulation the tips of two EG stalks swing like a pendulum with 28~50 Å movement. In our study the suggested circular motion of EG is modest with ~ 20 Å movement.

2-2 reviewer's suggestion

The text seems a little confusing, in that the stalks are “rigid and inflexible” (line 318), but “bend” at the same time (line 191). I feel this point should be clarified and discussed.

2-2 Our response

We agree with the referee’s comment. Accordingly, the sentences were modified as follows;

Line181- These comparisons clearly indicate that the stalk regions of both EG1 and EG2 are asymmetrical during rotation of the central rotor, and this is likely coupled to motion of the α -NT relative to the α -CT.

Line306- The two EG stalks have the same structures in the current model, both adopting conformations that match the crystal structure²⁷, while their conformations in the complex are different when superimposing the β -barrel domain of the B-subunit (Fig. 3b).

2-3 reviewer’s suggestion

(ii) lines 273-274. “This is the first report of nucleotides bound to the rotary ATPase in the native frozen state.” and 2-7 in the response “Our study reveals the first EM structure of a rotary ATPases with nucleotides bound in the catalytic sites.” □ This statement isn’t correct, Sobti et al. 2016 (reference 17) clearly identify nucleotides in their map of E. coli ATP synthase (see Figure 5). The text should be edited accordingly.

2-3 our response

As mentioned by the reviewer, the sentence “This is the first report of” is not appropriate. Thus we remove the sentence in the text.